# Genome Sequencing of five *Lacticaseibacillus* Strains and Analysis of Type I and II Toxin-Antitoxin System Distribution

**DOI:** 10.3390/microorganisms9030648

**Published:** 2021-03-21

**Authors:** Alessia Levante, Camilla Lazzi, Giannis Vatsellas, Dimitris Chatzopoulos, Vasilis S. Dionellis, Periklis Makrythanasis, Erasmo Neviani, Claudia Folli

**Affiliations:** 1Department of Food and Drug, University of Parma, 43124 Parma, Italy; camilla.lazzi@unipr.it (C.L.); erasmo.neviani@unipr.it (E.N.); 2Biomedical Research Foundation of the Academy of Athens (BRFAA), 115 27 Athens, Greece; gvatsellas@bioacademy.gr (G.V.); dchatzop@bioacademy.gr (D.C.); vasileios-stamatios.dionellis@unige.ch (V.S.D.); pmakrythanasis@bioacademy.gr (P.M.); 3Department of Molecular Biology, University of Geneva, 1211 Geneva, Switzerland

**Keywords:** *Lacticaseibacillus*, genome sequencing, toxin-antitoxin systems, Lpt toxin, DinJ/YafQ, MazEF, YefM/YoeB, Phd/Doc

## Abstract

The analysis of bacterial genomes is a potent tool to investigate the distribution of specific traits related to the ability of surviving in particular environments. Among the traits associated with the adaptation to hostile conditions, toxin–antitoxin (TA) systems have recently gained attention in lactic acid bacteria. In this work, genome sequences of *Lacticaseibacillus* strains of dairy origin were compared, focusing on the distribution of type I TA systems homologous to Lpt/RNAII and of the most common type II TA systems. A high number of TA systems have been identified spread in all the analyzed strains, with type I TA systems mainly located on plasmid DNA. The type II TA systems identified in these strains highlight the diversity of encoded toxins and antitoxins and their organization. This study opens future perspectives on the use of genomic data as a resource for the study of TA systems distribution and prevalence in microorganisms of industrial relevance.

## 1. Introduction

Toxin–antitoxin (TA) systems are widely distributed in bacterial genomes and are involved in various cellular processes, including adaptive response to environmental conditions, by regulating cell growth and/or death [1,2]. TA systems generally consist of a toxin that interferes with essential cellular functions, and a cognate antitoxin, capable of neutralizing the toxic effect. TA systems are classified according to the nature of the antitoxin and its mode of inhibition of toxin activity [3]. In particular, type I TA systems are composed of a toxic peptide and a non-coding RNA antitoxin able to inhibit toxin expression by interacting with toxin encoding mRNA. In type II TA systems, both toxin and antitoxin are proteins that can interact forming an inactive complex. Type I TA systems located in plasmid DNA are involved in plasmid maintenance, while the role of their homologs identified in chromosomal DNA has not been clarified. Type II TA systems, usually identified in chromosomal DNA, represent bacterial stress response mechanisms. Currently, the increasing availability of genome sequences has allowed us to grasp more about the complexity of bacterial lifestyle and adaptation features and has expanded the possibilities to identify and describe new TA systems. Attention has been given to the distribution and prevalence of TA systems in various food-associated microorganisms. Recently, genome-wide screening allowed the identification of novel TA systems in the food-borne pathogens *Staphylococcus aureus* [4] and *Listeria monocytogenes* [5]. TA systems are widespread also in the genomes of strains relevant for industrial productions, such as lactic acid bacteria (LAB), but still poorly studied. In the case of LAB, the activation of TA systems might modulate bacterial adaptation to processing stresses and might contribute to the adaptation to various niches, such as dairy products or the gastrointestinal tract of vertebrates [6,7,8].

Bacterial species belonging to *Lacticaseibacillus* genus are characterized by phylogenetic and metabolic diversity [9] and their ecological niche plays usually an important role in defining their metabolic properties. Nevertheless, strains belonging to the species *Lacticaseibacillus casei*, *Lacticaseibacillus paracasei* and *Lacticaseibacillus rhamnosus* can be isolated from various environments and are referred to as nomadic, possessing traits for adaptation to diverse selective conditions [7]. In particular, members of these species share gene sets described as pan-genome in various genomic comparison studies [6,10,11] but a certain degree of variability is maintained, as observed in a large dataset of *L. rhamnosus* isolated in different niches [12]. In particular, dairy associated strains are characterized by the loss or the inactivation of sugar transport systems and mucous-binding pili encoding regions, and by a typical dairy signature of the CRISPR locus profile, possibly due to the ubiquitous phage predation observed in this niche. Similarly, dairy isolates belonging to the species *L. casei* and *L. paracasei* show the loss of genes involved in the metabolism of carbohydrates or non-essential amino acids in milk-derived products, while gene acquisitions are mostly related to plasmids and phage defense mechanisms [13,14].

Among the various adaptation traits, the presence of TA systems is poorly investigated, despite these systems can promote bacterial adaptation and survival to the varying environmental conditions encountered during the cheese-making process [15,16]. 

Recent works have identified the type I Lpt/RNAII TA system in members of *Lacticaseibacillus* and have characterized its functionality [17,18] In addition, a wide distribution of type II TA systems was revealed in the same genus [19,20], and the regulation of some of these systems in response to stress conditions related to food manufacturing was investigated [16]. In this study, the genomes of five *Lacticaseibacillus* strains isolated from dairy products were analyzed and compared according to their genomic features with particular attention to type I and type II TA systems distribution. 

## 2. Materials and Methods

### 2.1. Bacterial Strains and Culture Conditions

*L. rhamnosus* (strains 1019, 1473) and *L. paracasei* (strains 2333, 4186 and 2306), belonging to the University of Parma Culture Collection (UPCC), were isolated from dairy matrices. Bacterial strains were maintained as frozen stocks (−80 °C) in Man Rogosa Sharpe (MRS) medium (Oxoid, Milan, Italy) supplemented with glycerol 15% (*w*/*v*). The cultures were propagated three times with a 2% (*v*/*v*) inoculum in 6 mL MRS in sterile glass tubes and incubated in anaerobiosis (AnaeroGen, Oxoid, Milan, Italy) at 37 °C for 15 h. 

### 2.2. Nucleic Acid Extraction, NGS Library Preparation and Sequencing

Total DNA extraction was performed on 1 mL of overnight cultures prepared as described above using DNeasy Blood & Tissue kit (Qiagen, Milan, Italy), following manufacturer’s instructions for lysis of Gram+ bacterial cells. Plasmid DNA extraction was performed on 1 mL overnight cultures with Plasmid Mini Prep kit (Fisher Molecular Biology, Rome), following the manufacturer’s instructions. An amount of 1 ng DNA from each sample was used to prepare Next Generation Sequencing libraries with the Illumina Nextera XT library prep kit. Quality control of the libraries was performed with Agilent Bioanalyzer HS DNA kit and quantitation with Qubit DNA HS kit. Approximately 4 Million 2x150 bp PE reads were generated for each sample in Illumina MiSeq Sequencer.

### 2.3. Genome Assembly, Annotation and Bioinformatics Analysis

Illumina sequence de novo assembly was performed using a Unicycler [21], and the quality of the assembled genomes was evaluated by the use of QUAST software [22]. The presence of rearrangements in the newly sequenced genomes was evaluated by comparison with reference strains isolated from dairy products, i.e., *L. rhamnosus* LC705 for *L. rhamnosus* 1019 and 1473 strains, and *L. paracasei* JCM8310 for *L. paracasei* 2333, 4186 and 2306. A circular representation of the sequence rearrangements was drawn with Circos software [23]. Annotation of the assembled genomes was performed by RAST 2.0 server (https://rast.nmpdr.org/rast.cgi, accessed on 14 February 2019). 

Pan-genome analysis was performed on the newly sequenced *Lacticaseibacillus* strains and 15 available reference strain genomes by using Roary software with default parameters [24] (Appendix A). Core gene nucleotide sequences alignments generated by Roary were used to obtain a maximum-likelihood phylogenetic tree using RAxML [25]. Definition of contigs as plasmid sequences was performed according to a four-step method: (i) identification of plasmid-derived contigs by mlplasmids (v. 1.0.0, https://sarredondo.shinyapps.io/mlplasmids/, accessed on 5 May 2020); (ii) identification of matching plasmid sequences with plasmidfinder (v. 2.1, https://cge.cbs.dtu.dk/services/PlasmidFinder/, accessed on 11 May 2020); (iii) encoding of Type I TA systems; (iv) identity above 95% with plasmid sequences hits deposited in the National Center for Biotechnology Information (NCBI) database (https://blast.ncbi.nlm.nih.gov/, accessed on 11 May 2020). Functional annotation of plasmid coding sequences was performed by using BlastKOALA [26] (https://www.kegg.jp/blastkoala/, accessed on 30 May 2020).

Manual annotation of type I TA systems homologous to Lpt/RNAII [17] was performed by a tBLASTn search using the Lpt amino acid sequence as a query. The prediction of putative promoter and transcription terminator sequences was carried out with the BPROM (http://www.softberry.com, accessed on 1 March 2021) and with the “Arnold finding terminators” web services (http://rna.igmors.u-psud.fr/toolbox/arnold/, accessed on 1 March 2021), respectively. 

Type II TA systems were initially annotated by RAST software. In addition, a manual BLASTn search by using *dinJ/yafQ* and *yefM/yoeB* identified in Ferrari et al. [19] or *mazEF* and *phd/doc* sequences deposited in the databases led to the identification of unannotated sequences. 

Multiple sequence alignments were constructed by using Clustal Omega and rendered with the GeneDoc program [27]. This Whole Genome Shotgun project was deposited at GenBank under the following accession numbers: *L. rhamnosus* 1019, JAFEHI000000000; *L. rhamnosus* 1473, JAFEHJ000000000; *L. paracasei* 2333, JAFEHK000000000, *L. paracasei* 4186, JAFEHL000000000 *and L. paracasei* 2306, JAFEHM000000000.

## 3. Results

### 3.1. Genomic Features of the Lacticaseibacillus Isolates

Assembly and annotation were performed on the genomes of the five *Lacticaseibacillus* isolates. Details about genome size, number of contigs and features for each strain are reported in Table 1. The genome of *L. paracasei* 4186 is the smallest one (2,922,588 bp), while the strain *L. paracasei* 2306 shows the largest genome (3,240,492 bp). The number of annotated coding sequences ranges from 3232 in *L. rhamnosus* 1019 to 3915 in *L. paracasei* 2333. In the genomes of the newly sequenced microorganisms, 26 contigs can be attributed to plasmids, according to the analyses performed by using dedicated software and data available in public repositories (Table 2). For each strain, a number of plasmids ranging between three (*L. paracasei* 4186) and eight (*L. paracasei* 2333) was identified. The largest plasmid, p1019-1 containing 27,278 bp, was identified in *L. rhamnosus* 1019, while the smallest one, p2333-8, 2076 bp long, was found in *L. paracasei* 2333. Of the 318 coding sequences located on the retrieved plasmids, only 43 could be associated to known functions (Appendix A). Most of the newly detected genes are related to plasmid maintenance functions, but also genes involved in amino acid, sugar and nucleotide metabolism, or related to antimicrobial resistance have been identified. Among these traits, plasmid p1019-1 from strain *L. rhamnosus* 1019 possesses a gene cluster, encoding the enzymes CysE2 and CysK2, respectively, a serine O-acetyltransferase and a cysteine synthase, and for the enzyme Ctl1, a cysteine-S-conjugate beta-lyase (Appendix A). This cluster is conserved in strains *L. paracasei* 2306 and *L. paracasei* 4186, with amino acid sequence identities above 99%. Interestingly, the sequences coding for the enzymes CysK2 and Ctl1 are extremely conserved also in strains *Lactobacillus delbrueckii* sp. *bulgaricus* American Type Culture Collection (ATCC) 11842 and *Streptococcus thermophilus* CNRZ1066 (percentage of identity ≥ 97%). Conversely, a greater divergence is observed for CysE2 coding sequence, which is absent in *L. delbrueckii* sp. *bulgaricus* ATCC 11842, while *S. thermophilus* CNRZ1066 gene shares an identity of 59% with *Lacticaseibacillus* sequences (Appendix A).

Each newly sequenced genome was compared with a strain of the same species isolated from a dairy product and available in the public database, that was chosen as a reference sequence. This comparison highlights different levels of sequence rearrangements (Figure 1). In particular, comparing *L. rhamnosus* genomes with that of the strain LC705 used as a reference sequence, we can observe only a few rearrangements for strain 1019 but several structural variations for strain 1473. In the case of *L. paracasei* genomes, numerous structural variations are observed when aligned with the selected reference strain, *L. paracasei* JCM8130. 

A pan-genome reconstruction was carried out, comparing the 5 isolates described in this study with 15 reference strains available in public databases (Appendix A). Reconstruction of phylogenetic relationship according to core gene alignment shows that strains belonging to the *L. rhamnosus* species form a distinct clade, comprising *L. rhamnosus* 1019 and 1473 strains (Figure 2a). Strains belonging to the species *L. casei* and *L. paracasei* form a single group, except for the strains *L. casei* ATCC 393 and *L. casei* LC5, that form a separate, small clade. The strains *L. paracasei* 2333 and *L. paracasei* 4186 appear to be closely related, while strain *L. paracasei* 2306 has a greater distance from these two strains (Figure 2a). The latter strain was previously reported as belonging to the species *L. casei* [28], but genome-scale comparison allowed correct attribution of strain 2306 to the species *L. paracasei*. Pan-genome analysis reported in Figure 2b was conducted on a set of 10,020 genes for all the 20 *Lacticaseibacillus* strains. Of these, 822 are shared by at least 19 strains, representing the core genome. Of the remaining genes, 4421 are shared by 3 to 18 strains, while 4777 genes were unique or shared only by two strains (Figure 2b). 

Considering the dairy origin of the five newly sequenced *Lacticaseibacillus* strains, a comparison was made among them, to highlight common and unique genes (Figure 2c). A total of 862 genes shared among all the analyzed strains were identified, mostly coding for metabolic traits such as carbohydrates utilization, amino acid and protein metabolism (data not shown). This result is in line with data provided in the broader *Lacticaseibacillus* comparison (Figure 2b). The highest number of shared distinctive genes (1384) are identified in *L. rhamnosus* 1019 and 1473 as expected on the basis of the phylogenetic relationship. Similarly, strain *L. paracasei* 2333 and *L. paracasei* 4186 share 1196 genes that are not present in the genome of the other strains. The strain *L. paracasei* 2306 is the most diverse. It accounts 448 unique gene sequences and shares 183 and 127 distinctive genes with *L. paracasei* 2333 and *L. paracasei* 4186, respectively.

### 3.2. Type I TA Systems Distribution and Structure

Type I TA systems were predicted in the sequenced genomes by a tBlastn search using the aminoacid sequences of Lpt toxin and its homologs identified in Folli et al. [17] as queries.

Overall, 14 sequences encoding peptides homologous to Lpt were predicted spread in all the five analyzed strains, and most of the sequences are located in contigs putatively identified as plasmids, with plasmid p1473-1 encoding for two different peptides. In particular, four distinct putative toxins homologous to Lpt were found in both *L. rhamnosus* strains, three sequences in *L. paracasei* 2333, two sequences in *L. paracasei* 2306 and only one in *L. paracasei* 4186 (Table 3).

The multiple alignment of the corresponding amino acid sequences highlighted three distinct sequences, named Lptlike, Lptlike1 and Lptlike2, in addition to Lpt peptide (Table 3 and Figure 3). A single copy of Lpt peptide was identified in *L. paracasei* 2333 and 4186 and in *L. rhamnosus* 1019, while two distinct sequences were found in the strain 1473 of *L. rhamnosus.* In addition, four identical Lptlike, with 68% of sequence identity with Lpt, were identified in *L. rhamnosus* 1473 and 1019 (two sequences) and in *L. paracasei* 2306, while four identical Lptlike1, with 65% of sequence identity with Lpt, were found in *L. rhamnosus* 1019 and 1473, in *L. paracasei* 2306 and in *L. paracasei* 2333. Both these putative toxin peptides have been previously identified in *Lacticaseibacillus* plasmid sequences deposited in the DNA database [18]. Interestingly, an additional 30 amino acid peptides with a sequence identity of 43% with Lpt were recognized in *L. paracasei* 2333 (Lptlike2). The comparison of the amino acid sequences of the different peptides highlights a conserved hydrophobic stretch of nine amino acids, which could locate inside the cellular membrane based on the structural prediction of the Lpt toxin [27].

To evaluate the functionality of these putative toxins, nucleotide sequences located upstream and downstream of their coding sequences were analyzed to identify promoter and terminator regions capable of controlling toxin-encoding and antitoxin RNA transcription. Appendix A shows that these DNA regions are highly conserved among all the strains. Nevertheless, in the strain *L. rhamnosus* 1019 it was not possible to complete this analysis for a copy of Lptlike and for Lptlike1 because the contigs containing the toxin encoding regions do not encompass the entire TA loci. In particular, Lptlike1 contig interrupted 48 nt after the toxin stop codon, while the contig encompassing Lptlike (named Lptlike Lr1019_1 in Appendix A), started 1 nt before the -10 sequence of the toxin promoter. In all the other putative TA systems, promoters have been correctly identified, with the exception of Lpt-Lr1473_1 of *L. rhamnosus* 1473 and Lptlike2 of *L. paracasei* 2333. In particular, in *L. rhamnosus* 1473 it was not possible to predict functional promoters capable of controlling Lpt RNA and antitoxin RNA synthesis, suggesting that the TA system is inactive. In *L. paracasei* 2333, a putative functional promoter is located upstream of the Lptlike2 coding sequence while no promoter sequences were identified on the complementary strand able to control antitoxin synthesis, suggesting that the toxicity of the peptide has been lost or that it could be controlled by a different mechanism of inhibition. An additional feature typical of type I TA systems are the direct repeat sequences located on toxin-encoding and antitoxin RNAs. These complementary sequences are involved in the RNA–RNA interaction inducing inhibition of toxin activity. In all the systems potentially active, these regions have been identified on both RNAs and are indicated in Appendix A. A graphical summary of the identified Lpt/RNAII systems is reported in Figure 4.

### 3.3. Type II TA Systems Distribution and Characteristics

Automatic annotation of the newly sequenced genomes by using the RAST server allowed us to find toxin and antitoxin sequences attributed to type II TA systems. A BLAST manual search by using the sequences described in the Materials and Methods section led to the identification of additional sequences related to type II TA systems (Figure 5 and Figure 6). It is interesting to note that each isolate encodes several toxin and antitoxin sequences but, not all are organized in toxin/antitoxin pairs, therefore some sequences are here categorized as orphans (Figure 5). The most represented TA systems are DinJ/YafQ and MazEF, classified as ribosome-dependent and ribosome–independent RNA interferase, which are spread in all the analyzed strains in single or multiple copies. In addition to these systems, complete or incomplete YefM/YoeB and Phd/Doc systems were identified in different strains (Figure 5 and Figure 6). 

About the DinJ/YafQ TA system, eight complete homologs were identified spread in all the strains, while orphan toxins were found in all the analyzed genomes except *L. rhamnosus* 1019 (Figure 5 and Figure 6). The alignment of all the identified YafQ sequences is reported in Figure 6a. The four orphan toxins (YafQ_pa2333_1, YafQ_pa2306_3, YafQ_pa4186_1 and YafQ_rh1473_2) share high sequence identity ranging from 72 to 100% but are quite dissimilar from the other identified YafQ proteins (sequence identity from 14 to 27%) (Appendix A). Among YafQ proteins belonging to complete systems, four toxins (YafQ_pa4186, YafQ_pa2333, YafQ_pa2306_1, YafQ_rh1473_1) share high sequence identity (from 76 to 98%) (Appendix A), while two YafQ homologs found in *L. paracasei* 2306 (YafQ_pa2306 and YafQ_pa2306_2), show sequence identity ranging from 11% to 38% with the other YafQ toxins. In addition, the truncated YafQ form, lacking the amino-terminal end previously characterized by Ferrari et al. [19], was found in both the *L. rhamnosus* genomes (YafQ_rh1473 and YafQ_rh1019) (Appendix A).

From a functional point of view, it is interesting to note that the truncated form of YafQ (YafQ_rh1473 and YafQ_rh1019) is inactive [19] and that the orphan YafQ proteins (YafQ_pa2333_1, YafQ_pa2306_3, YafQ_pa4186_1 and YafQ_rh1473_2) are characterized by two mutations (D61/A and D67/N or S) (Figure 6a) involving two out of the four catalytic residues identified in *E. coli*, suggesting their inactivation. Data reported in the literature on *E. coli* YafQ have indeed shown that the mutant YafQ D61/A maintains 50% of the enzymatic activity, while the replacement of aspartate 67 with alanine leads to the loss of toxicity [29]. 

Comparing *E.coli* YafQ catalytic residues (H50, H63, H87, D61, D67 and F91) with those found in the YafQ proteins belonging to complete systems of *Lacticaseibacillus* strains, it is possible to note that the three histidines are highly conserved with the only exception of YafQ_pa2306_1, in which H63 has been substituted with a tyrosine. D61 is conserved or replaced by an E residue except in YafQ_rh1473_1 in which is substituted with a K, while D67 is replaced by a G in YafQ_pa2333 (Figure 6a). This latter mutation induces the loss of toxic activity as reported in Ferrari et al. [19].

The identified DinJ antitoxins share similar trends in terms of amino acid composition. DinJ_pa4186, DinJ_pa2333, DinJ_pa2306_1, DinJ_rh1473_1 share sequence identity in the range of 74%-100%, while DinJ_pa2306_2 and DinJ_pa2306 appear more dissimilar from the other DinJ homologous sequences (sequence identity 4–24%) (Appendix A).

In the case of MazEF, a single copy was identified in each strain. The identified MazF toxins share a high sequence identity, ranging from 98% to 100%, and show the conservation of both the catalytic residues involved in the RNA cleavage (Figure 6b) [30]. MazE antitoxins present more variability among the strains, in a species-dependent way, and show 86% of sequence identity between *L. paracasei* and *L. rhamnosus* (Appendix A). 

In the case of YefM/YoeB, the antitoxin YefM was identified in all strains except for *L. paracasei* 2333, while only two YoeB toxins were found in the two *L. rhamnosus* strains (Figure 5). These YoeB proteins show a sequence identity of 97% and contain both the catalytic residues identified in *E. coli* (Figure 6c) [31]. The YefM antitoxins have sequence identities ranging from 35% of the truncated YefM_pa2306 sequence, lacking the amino-terminal end, up to 98% (Appendix A).

Finally, two complete Phd/Doc systems and four orphan antitoxins Phd were identified. In the Phd/Doc pairs of *L. paracasei* strains 2333 and 4186, toxins have identical amino acid sequences and show the typical catalytic motif HXFX(D/N)(A/G)NKR (Figure 6d), while their cognate antitoxins (Phd_pa2333 and Phd_pa4186) share a sequence identity of 98% (Appendix A). The orphan antitoxin Phd_pa2333_1 is shown as incomplete because it was not possible to identify its N-terminal aminoacid sequence due to its location at the beginning of the contig. Among the other orphan Phd sequences, Phd_rh1473 and Phd_rh1019_1 share an identity of 95%, while Phd_rh1019 is very dissimilar, showing a sequence identity of 8% and 9% with Phd_rh1473 and Phd_rh1019_1, respectively (Appendix A).

A preliminary search aimed to identify type II TA systems in the strains used as references in the genome analysis (Appendix A) led to the identification of several complete systems. MazEF and DinJ/YafQ were the most spread among the analyzed strains with one copy of MazEF for each strains and five copies of DinJ/YafQ found in *L. casei* ATCC393 and in the strains BD-II, BL23, JCM8130 and W56 of *L.paracasei*. In addition, three complete YefM/YoeB were identified in *L. rhamnosus* (ATCC 8530, Lc 705 and LOCK908 strains) and one complete Phd/Doc system was found in the strains JCM8130 of *L. paracasei*.

## 4. Discussion

Comparative analysis of microbial genomes represents an unprecedented tool for phylogenetic analysis and allows the provision of insights on strain-level diversity and niche adaptation [7]. The genus *Lacticaseibacillus* was recently proposed to include the species formerly known as *L. casei*, *L. paracasei* and *L. rhamnosus*, which form a clade of closely phylogenetically related microorganisms with relevant application in the food industry [9,11]. Bacteria from this genus are isolated from a variety of habitats, among which are dairy products, where they can contribute to the development of cheese aromatic properties [32]. Specific traits related to the adaptation to this environment have been identified in some strains [13].

Isolates of the *Lacticaseibacillus* group are often reported as nomadic lactobacilli, as a reference to their frequency of isolation from a wide variety of niches [7]. In the perspective of a nomadic lifestyle, plasmids can carry several genes important for the growth and adaptation of microorganisms to varying environmental conditions, as well as technological traits useful for the selection of strains for industrial applications [33]. Indeed, a comparative analysis of plasmids from the broad *Lactobacillus* genus corroborated the association between plasmids and functional traits involved in niche adaptation, highlighting the enrichment of genes associated with transposition, and defense systems such as restriction–modification enzymes and toxin–antitoxin systems [34].

Plasmid analysis performed in this study has revealed a small cluster of genes involved in the metabolism of the sulfur-containing amino acids cysteine and methionine, encoded in plasmid p1019-1 found in *L. rhamnosus* 1019. Furthermore, the same cluster is present also in the genomes of the strains *L. paracasei* 2306 and *L. paracasei* 2333. Utilization of sulfur-containing amino acids is linked to the production of key flavor compounds, representing a trait of interest for the selection of strains for the food industry [35]. The cluster identified in this study was recently described in a paper, which investigated the transcriptional regulation of cysteine and methionine metabolism in *L. paracasei* FAM18149, reporting the overexpression of this operon in a chemically defined media omitting cysteine or methionine [36]. A previous work reported the presence of genes involved in methionine and cysteine metabolism in the plasmid pLC1 from *L. rhamnosus* Lc705, suggesting the occurrence of plasmid-mediated transfer from species like *S. thermophilus*, *L. helveticus* and *L. delbrueckii* sp. *bulgaricus* [37]. It is noteworthy that all the *Lacticaseibacillus* strains included in the present work were isolated from dairy products, which is a common source of isolation for bacteria from this genus, highlighting their potential implication in the development of sensory attributes of dairy products [38,39]. 

A typical trait associated to the adaptation to hostile environments is the presence of toxin–antitoxin systems in bacterial genomes. In this work, we investigated the distribution of the main type II TA systems and of Lpt/RNAII homologs, a type I TA system previously identified in plasmid DNA of *Lacticaseibacillus* strains. The distribution of type I TA systems in the analyzed strains has shown their frequent location on plasmid DNA, as well as in multiple copies in agreement with the proposed role in the post-segregation killing phenomena [3]. Three toxin peptides, named Lptlike, Lptlike1 and Ltlike2, in addition to Lpt have been predicted but in vivo studies should be carried out to verify their toxicity; also, expression studies conducted on *Lacticaseibacillus* strains could provide more insight into the activity of these TA systems. Interestingly, the presence of type I TA systems loci on plasmid carrying genes involved in the expression of favorable properties, such as the production of volatile compounds, can promote the maintenance of the plasmid and the consequent preservation of the positive character. At the same time, most of the protein-coding sequences of the plasmids carrying type I TA systems are of unknown function; thus, future research could aim at their functional characterization.

The prevalence of type II TA systems is in agreement with the distribution of these systems in the *Lactobacillus* sequences deposited in the public databases, as reported in Ferrari et al. [19]. Different orphan YafQ toxins and orphan Phd/YefM antitoxins have been also identified. Interestingly, all the YafQ toxins not associated with the cognate antitoxins DinJ show the replacement of catalytic residues essential for their activity. On the contrary, the orphan antitoxins often share a high sequence identity with similar antitoxins associated with the cognate toxins suggesting that they could play a role in cross-regulation mechanisms [40]. The wide distribution observed for type II TA systems in these bacterial strains isolated from dairy products suggests an important role in the adaptation of these strains to environmental stresses, such as scarcity of nutrients typical of cheese. Recent studies have actually shown that the expression of DinJ/YafQ TA systems can be related to various environmental stresses, including those encountered in food production [16,41]. 

The strains sequenced in this work were previously characterized for their potential application as adjunct starters in dairy products [42], or for use in the production of fermented fruit juices [8]. It is noteworthy that strain *L. paracasei* 2306, that showed the highest number of identified type II TA systems, shows a phenotype of reduced growth in vegetable substrates, but retains metabolic activity associated with increased volatile compounds and modification of the polyphenolic profile [8]. Despite the fact that no association can be made between the presence and activation of TA systems, transcriptomic analysis could provide information on the activity of TA systems in different environmental conditions useful for finding interesting correlations with specific traits of interest for various industrial applications.

Although the biological functions of many TA modules and their role in the adaptation of microorganisms to various substrates have not been completely clarified, genome analysis can provide useful insights. By comparing the genomes of *Lacticaseibacillus* isolates from dairy matrices, it is possible to identify traits that are involved in the adaptation to the environment of cheese or in the development of the sensory profile of the final product. Moreover, the knowledge of TA systems in industrially relevant microorganisms is still limited, opening the possibility to shed light on the mechanisms that lead to the assembly of food microbiota.

## Figures and Tables

**Figure 1 microorganisms-09-00648-f001:**
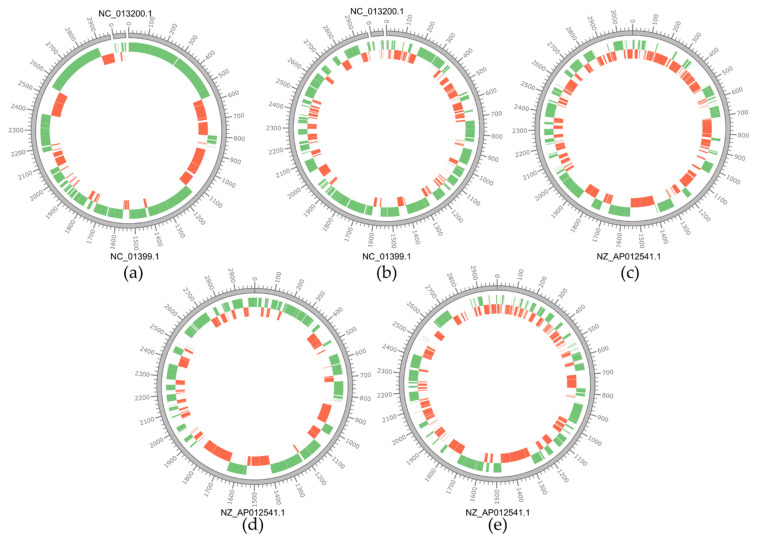
Circular representations of the chromosomes of *Lacticaseibacillus* isolates. The outer grey circle represents the genome of the reference strain used for the alignment, the green circle represents the contigs aligned with the reference sequence, the red circle represents structural contig rearrangements compared to the reference sequence. Genome scale is in Kbp. Below each circular representation the RefSeq accession number for each type strain is reported (see Appendix A for details). For strain L. rhamnosus Lc705 plasmid pLC1 was included in the comparison (RefSeq: NC_013200.1). (**a**) *L. rhamnosus* 1019 vs. *L. rhamnosus* Lc705; (**b**) *L. rhamnosus* 1473 vs. *L. rhamnosus* Lc705; (**c**) *L. paracasei* 2306 vs. *L. paracasei* JCM8130; (**d**) *L. paracasei* 2333 vs. *L. paracasei* JCM8130; (**e**) *L. paracasei* 4186 vs. *L. paracasei* JCM8130.

**Figure 2 microorganisms-09-00648-f002:**
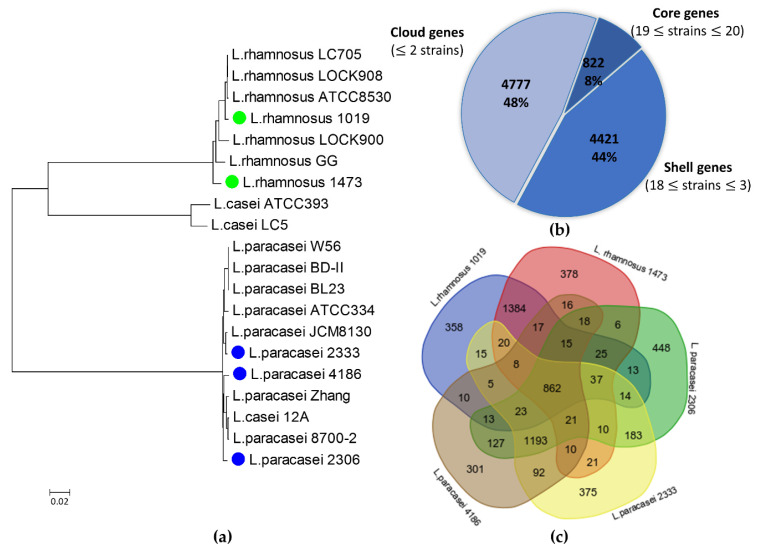
Phylogenetic analysis of *Lacticaseibacillus* isolates: (**a**) phylogenetic tree constructed by using newly sequenced strains and 15 reference strains, as inferred from core gene sequence alignment. Sequences were aligned using Roary, RAxML was used to construct a maximum-likelihood phylogenetic tree; (**b**) description of the pan genome of the 20 strains belonging to *Lacticaseibacillus* genus; (**c**) Venn diagram showing the number of shared and unique genes among the newly sequenced strains of *Lacticaseibacillus* genus.

**Figure 3 microorganisms-09-00648-f003:**
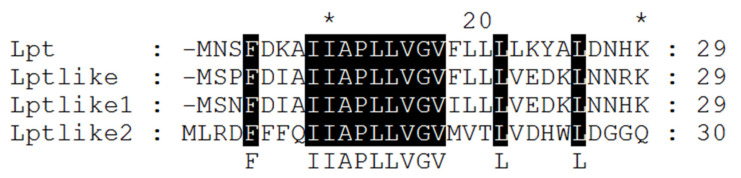
Multiple sequence alignment of the four different peptides identified in the analyzed strains. * 10 bp intervals along with the sequence length indicators

**Figure 4 microorganisms-09-00648-f004:**
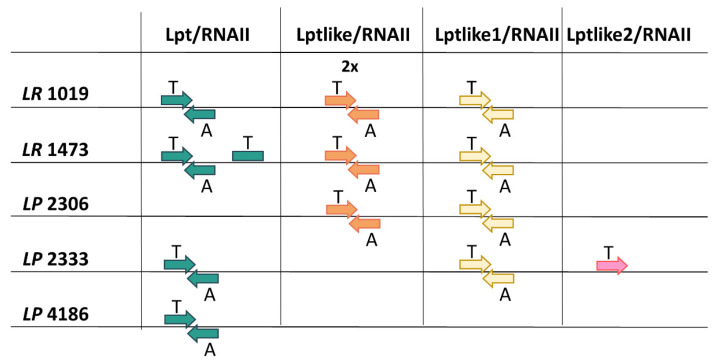
Distribution and organization of the identified type I TA systems. Toxins and antitoxins are indicated by T and A, respectively. Abbreviations in strain names are as follows: LP: *L. paracasei*, LR: *L. rhamnosus*. The arrows indicate transcription direction of T and A, their absence highlights the promoter Scheme 15. *Lactocaseibacillus* strains used as references for genome analysis (Appendix A). By using the tBlastn program, one Lpt and two Lptlike2 were identified, while no Lptlike and Lptlike1 were found. The identified Lpt sequence is located on the plasmid pLBPC-2 of *L. paracasei* JCM 8130 and it was previously described in Folli et al. [17]. Lptlike2 sequences were found in *L. paracasei* JCM 8130 and in *L. paracasei* ATCC334 and the alignment of the nucleotide sequences encompassing their coding regions is shown in Appendix A in comparison with Lptlike2 of *L.paracasei* 2333. Interestingly, it was not possible to predict convergently transcripted RNA molecules able to act as antitoxins in any of these sequences.

**Figure 5 microorganisms-09-00648-f005:**
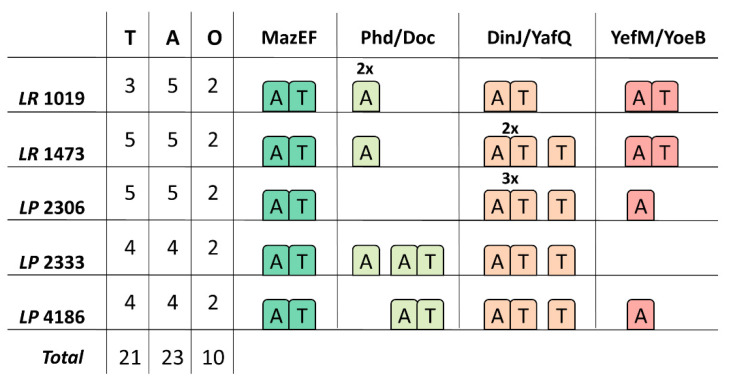
Distribution and organization of the identified type II TA systems. The left part of the figure summarizes the distribution of toxins (T), antitoxins (A) and number of TA systems not organized in pairs, that can be considered as “orphans” (O) in the *Lacticaseibacillus* isolates. The right part of the figure shows the organization of the identified TA systems in the strains. A superimposed number indicates that multiple copies were found in the genome. Abbreviations in strain names are as follows: LP: *L. paracasei*, LR: *L. rhamnosus*.

**Figure 6 microorganisms-09-00648-f006:**
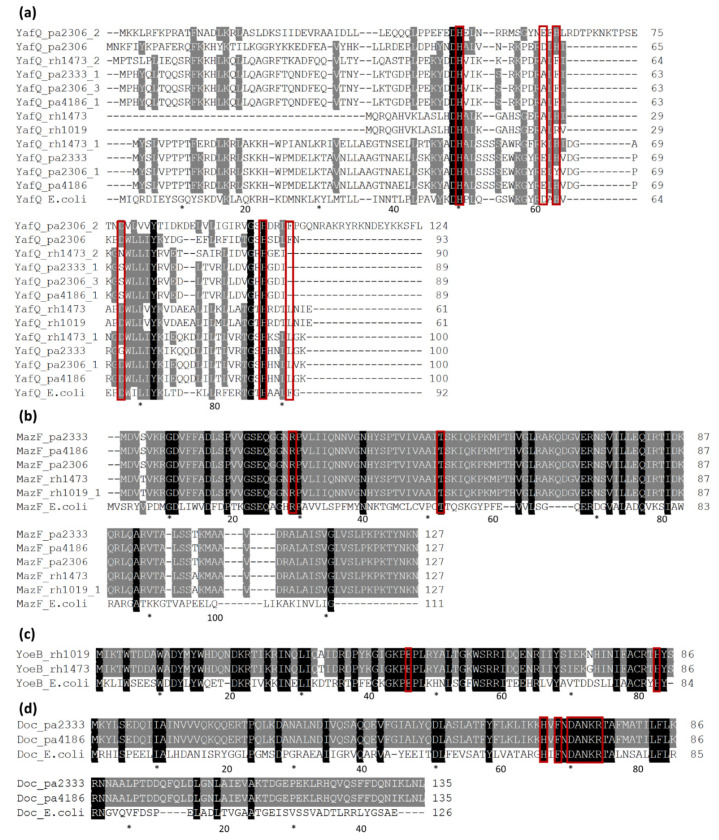
Multiple sequence alignments of all the toxins identified in *Lacticaseibacillus* strains compared with *E. coli* homologous sequences. (**a**) YafQ proteins, *E.coli* sequence accession number HAM8829092; (**b**) MazF proteins, *E.coli* sequence accession number WP_000254738; (**c**) YoeB proteins, *E.coli* sequence accession number WP_112934640; (**d**) Doc proteins, *E.coli* sequence accession number WP_001216045. The catalytic residues identified in *E. coli* proteins are boxed in red and the numbering of *E. coli* residues is reported. Abbreviations prior to strain names are: pa: *L. paracasei*, rh: *L. rhamnosus*. * 10 bp intervals along with the sequence length indicators

**Table 1 microorganisms-09-00648-t001:** Characteristics of the sequenced *Lacticaseibacillus* genomes. cds, protein coding sequences; ncRNAs, non-coding RNAs; ORF, open reading frame.

Strain	Isolation Source ^1^	Ripening (months)	Size (bp)	GC Content (%)	n° of Contigs	n° of ORF	cds	ncRNAs
*L. rhamnosus* 1019	PR	4	3,088,347	46.58	248	3300	3232	68
*L. rhamnosus* 1473	PR	20	3,088,357	46.57	384	3464	3402	62
*L. paracasei* 2306	PR	6	3,240,492	46.12	364	3637	3572	65
*L. paracasei* 2333	PT	n.a.	3,163,164	46.54	993	3979	3915	64
*L. paracasei* 4186	PR	1	2,922,588	46.27	396	3326	3264	62

^1^ Isolation source: PR, Parmigiano Reggiano cheese; PT, Pecorino Toscano cheese; n.a., not applicable.

**Table 2 microorganisms-09-00648-t002:** Characteristics of the newly identified plasmid sequences. Plasmid names were assigned according to the *Lacticaseibacillus* strain the plasmid belongs to. Contig length and coverage are reported, as well as the identification criteria. mlplasmids software (mL), plasmid finder suite (pf), encoding type I toxin–antitoxin (TA) systems homologous to Lpt (ta), accession number of plasmids identified in the National Center for Biotechnology Information (NCBI) database with sequence identity above 95% (plasmid).

Name	Length (bp)	Coverage (x)	mL	pf	ta	Plasmid
p1019-1	27,278	185	x	x		CP014202.1
p1019-2	4812	1487	x		x	CP001155.1
p1019-3	3955	177	x			CP048004.1
p1019-4	3644	1241	x			AY662331.1
p1019-5	2422	947			x	AP012543.1
p1473-1	15,540	584			xx	CP012190.1
p1473-2	14,433	121	x	x		CP001156.1
p1473-3	13,211	131		x		CP044229.1
p1473-4	8964	623	x		x	EU255257.1
p1473-5	5983	25			x	MH621333.1
p2306-1	10,022	164	x	x		CP044229.1
p2306-2	24,039	187		x		CP035564.1
p2306-3	8837	937			x	AY662330.1
p2306-4	5006	1682			x	EU255257.1
p2306-5	3223	141	x			CP048004.1
p2333-1	23,737	179	x			CP014885.1
p2333-2	11,765	709	x			CP035564.1
p2333-3	9153	618			x	Z50861.1
p2333-4	8630	191		x		CP044229.1
p2333-5	8002	615	x		x	AY662330.1
p2333-6	5455	95	x			CP035564.1
p2333-7	3897	150			x	CP007125.1
p2333-8	2076	160	x		x	CP044362.1
p4186-1	5673	30		x		CP007125.1
p4186-2	3824	114	x			CP012190.1
p4186-3	3097	153	x			CP017262.1

**Table 3 microorganisms-09-00648-t003:** Sequence encoding for Lpt toxins and its homologs in the *Lacticaseibacillus* strains. The corresponding plasmid contig is reported, when applicable. n.a., not applicable.

Strains	Lpt Toxins and Its Homologs	Location
*L. rhamnosus 1019*	Lpt_rh1019	p1019-5
	Lptlike_rh1019	p1019-2
	Lptlike_rh1019_1	n.a.
	Lptlike1_rh1019	n.a.
*L. rhamnosus 1473*	Lpt_rh1473	p1473-5
	Lpt_rh1473_1	p1473-1
	Lptlike_rh1473	p1473-4
	Lptlike1_rh1473	p1473-1
*L. paracasei 2306*	Lptlike_pa2306	p2306-4
	Lptlike1_pa2306	p2306-3
*L. paracasei 2333*	Lpt_pa2333	p2333-3
	Lptlike1_pa2333	p2333-5
	Lptlike2_pa2333	n.a.
*L. paracasei 4186*	Lpt_pa4186	n.a.

## Data Availability

The data presented in this study are openly available in GenBank (https://www.ncbi.nlm.nih.gov/genbank/, accessed on 10 March 2021) under the following accession numbers: *L. rhamnosus* 1019, JAFEHI000000000; *L. rhamnosus* 1473, JAFEHJ000000000; *L. paracasei* 2333, JAFEHK000000000, *L. paracasei* 4186, JAFEHL000000000 *and L. paracasei* 2306, JAFEHM000000000.

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
