# Peer review of "Genome Sequencing of five Lacticaseibacillus Strains and Analysis of Type I and II Toxin-Antitoxin System Distribution"

_microorganisms, 2021, doi:10.3390/microorganisms9030648_

Round 1

Reviewer 1 Report

In the study by Levante and colleagues, the genomes of five Lacticaseibacillus strains, isolated from dairy products, were sequenced and analyzed, with a special focus on type I and II TA systems. Homologs of the type I TA system Lpt/RNAII were mainly found on extrachromosomal DNA (plasmids), and are likely involved in post-segregational killing. The type II TA systems MazEF and DinJ/YafQ were found in all isolates, while Phd/Doc and YefM/YoeB were only found in some. The type II TA systems likely contribute to stress adaptation in the natural environments of the isolates.

The present study is a continuation of former work on TA systems in lactic acid bacteria. The data are very descriptive, and detailed analyses are clearly needed in the future to shed light on the function of the discovered TA systems. However, identification of TA systems is a first important step.

The manuscript is mostly well-written. I have some comments that should help to improve presentation of the data and readability of the manuscript.

Broad comments

  1. The introduction would clearly benefit from a paragraph that explains type I and type II TA systems (what are the differences / similarities of these systems etc.).

  1. The caption of Table 3 is misleading since it refers to TA systems. However, only Lpt homologs (toxin) are reported. Please add information on RNAII homologs (antitoxin) to Table 3!

  1. Line 238ff: The paragraph on direct repeat (DR) sequences is not clear to me. When consulting Figure S2, I cannot easily recognize any direct repeat. Also, the DR sequence is only illustrated for what I believe is the RNA antitoxin. Figure S2 should be modified in a way that also transcription start sites and potential terminators are visible. This would help to identify the proposed interaction between the DR sequences.

  1. Line 256ff: Based on Figure 5, it is somewhat difficult to compare all the different YafQ homologs. It would be helpful to show separate alignments (at least in the supplement) for subgroups of YafQ toxins with strong homology.

  1. In the discussion, I am missing some information on future directions. Is it feasible to manipulate the Lacticaseibacillus strains to improve dairy products? Or should production conditions be modified to alter expression of TA systems?

Specific comments

Line 23, 102, 312ff, Figure S3: ‘PhD’ should read as ‘Phd’.

Line 35: Do you mean ‘Staphylococcus aureus’?

Line 44: ‘isolated by’ should read as ‘isolated from’.

Line 45: What do the authors mean by saying ‘adaptation to diverse selective’?

Line 69: Which culture volume / flasks did you use? Please add some more information on culture conditions!

Line 123, 165, 190: Skip the word ‘for’ in the term ‘encoding for’.

Line 124 (and later): Use capital letter for protein names (e.g., CysE2 and CysK2).

Line 136: ‘paracasaei’ should read as ‘paracasei’.

Line 163ff, Table 2: Please explain in the caption that the plasmid name indicates to which strain the corresponding plasmid belongs.

Line 181, Figure 2: The caption only refers to ‘phylogenetic analysis of L. casei’. What is about L. rhamnosus and L. paracasei?

Line 190ff: It might be better to call these sequences ‘Lpt homologs’. Whether these peptides are toxic or not remains to be demonstrated (as you also state in lines 358f).

Line 202, Table 3: Do you refer to Lactobacillus strains or rather Lacticaseibacillus strains?

Line 210f: The term ‘two sequences each’ is misleading, since it indicates that two Lptlike peptides were found in both L. rhamnosus 1473 and 1019.

Line 244: I am not sure whether ‘recover’ is the right word. Do you mean ‘discover’? Please also check line 103!

Line 291: Are all catalytic residues highlighted in Figure 5?

Line 266ff, Figure 4: It would be helpful to explain abbreviations ‘LR’ and ‘LP’ in the legend. Also, in the text TA systems are written as e.g. Phd-Doc, in the figure legend as Phd/Doc. Please conform to one style!

Line 268: Please check the spelling of Lacticaseibacillus!

Line 303f: What does ‘more variable according to the strain species’ exactly mean?

Line 315: Do you mean ‘amino-terminal end’?

Figure 1: Please improve the quality of the figure! One cannot even recognize whether there is any text to be read.

References: Please make sure that all species and gene names are italic.

Reviewer 2 Report

Mayor comments

  • Why the authors did not include their own Lacticaseibacillus casei strains in their study?
  • The authors should provide more functional information of the five studied strains: Are they good starters for yogurt, cheese or other dairy products? Are they used in specific dairy products productions? Do they have different metabolic activities like acidifying milk substrates or producing aromatic compounds? Are they able to grow equally in milk substrates? Can they confer different sensory attributes to dairy products? And more importantly, are any of these functional differences or similarities related to differences or similarities in the genomes found in this work (especially in the toxin-antitoxin system distribution)?
  • The authors have nicely summarized the main findings of their work in the figure 4, showing a distribution and organization of the identified Type II TA systems in their sequenced strains. They should presents a similar summary for the identified Type I TA systems. Furthermore, in addition to the 5 new sequenced strains, references strains of dairy and non-dairy origin should be included for comparison in both Type I and Type II analysis.

Minor comments

  • Authors should use the new template for manuscript of Microorganisms.
  • Please correct several space and typo errors in the manuscript. And also please avoid to repeat constantly “in the case”.
  • Please arrange the positions of tables and figures in the manuscript to facilitate the lecture of the results.
  • The quality of some figures like figure 1 and 5 needs to be improved.
  • Page 2, line 45. It is not clear what the authors meant with “adaptation to diverse selective”
  • Page 2, lines 65-66. Please specify the dairy matrices from which each strain was isolated.
  • Table 1. Why the numbers of contigs in the strain 2333 is significantly higher than the other strains?
  • Page 3, lines 97-101. Please correct the format of the paragraph.
  • Page 3, lines 106-107. Please indicate the strain that correspond to each accession number.
  • Page 3, lines 132-138. Please briefly explain the reasoning for selecting the LC705 and the JCM8130 strains as references and not other strains like the GG strain for example.

Reviewer 3 Report

The manuscript “Genome sequencing of five Lacticaseibacillus strains and analysis of type I and II toxin-antitoxin system distribution” analyses i) the genome of bacterial species belonging to Lacticaseibacillus genus and ii) the distribution of type I and II TA systems in these species.

This manuscript comprises:

1 – The assembly and annotation of diverse genome.

2 – Bioinformatic analyses to identify systems homologous to TA I and II within the Lacticaseibacillus genus.

Some specific experimental issues:

The analysis and sequence comparisons are performed correctly, but the main problem I have found in this test is the comparison between E. coli and Lactobacillus systems due to their phylogenetic distance. The authors support the homology found within the genus Lactobacillus with the LPT toxin and DinJ/YafQ system on their previously published data (Folli et al., 2017, Maggi et al., 2019 and Ferrari et al., 2019 respectively). These data provide strength to their comparisons, however when the authors compared with TA systems in E. coli, the degree of identity is rather low and it is not correlative. Moreover, this fitting is somehow forced because they tried to adjust the residues that conform the catalytic pocket of the toxin in the sequence alignment.

Line 372: “Although the biological functions of many TA modules and their role in the adaptation of microorganisms to various substrates have not been completely clarified, genome analysis can provide useful insights”. Authors show TA modules in the newly identified plasmid sequences, but they do not prove a direct relationship between the selection of TA genes with key genes for flavour compounds.

Line 374: “By comparing the genomes of Lacticaseibacillus isolates from dairy matrices, it is possible to identify traits that are involved in the adaptation to the environment of cheese and in the development of the sensory profile of the final product”. TA modules are associated with cell survival under different stress conditions: one part of the population altruistically sacrifices entering in a dormancy state, while another part of the population survives. When the stressful conditions become favorable, bacteria can activate their metabolism again. TA systems could be coded in the same CRISPR-CAS locus with other system that confers immune protection against phages or in plasmids together with antibiotic resistance genes. All these genes confer an advantage against other microorganisms. However, thinking that TA systems are associated with other genes in order to “improve the sensory profile of the food” is a very anthropological perspective.

The conclusions reached by the authors about the use of massive genome sequencing as a tool for selecting strains in industrial food seems to be a suitable approximation, since new strains that have acquired genes encoding for enzymes that improve the organoleptic properties could be identified in order to be selected later. Nevertheless, in the particular case of the genes that conform TA systems I do not find that this genome analysis could be relevant for searching strains with industrial interest.

Minor issues:

Figure 5: The catalytic residues highlighted in red are smudgy, it would be better to point them also with a box. The numbering of residues is hard to follow due to the high diversity among them. It would be interesting to highlight also the most significant residues that are described in the text.

Line 310: N-terminal aminoacid sequence by “aminoterminal aminoacid sequence”

Reference 19, Ferrari et al., 2019 is duplicated (ref 28).

Round 2

Reviewer 1 Report

I am pleased with the corrections and modifications of the manuscript. I only have two minor comments:

Line 35: Inhibition of toxin activity suggests protein-protein interaction (as in type II systems). I would rather say that RNA antitoxins inhibit toxin expression (or synthesis), since they interfere with ribosome binding and translation initiation.

Line 557f: What does ‘expression of positive characters’ mean? Rather delete the term (I don’t think the term is necessary for the point you want to make).

Author Response

Dear Reviewer #1,

We have modified the text as suggested, and we thank you again for your helpful comments.

Reviewer 2 Report

The authors improved the manuscript according to the suggestions and responded all the doubts.

I still believe that the quality of the figures can be improved further.

Author Response

Dear Reviewer #2,

We have improved the quality of figures 1 and 6, as suggested.